# RANDOMIZED FEATURE SQUEEZING AGAINST UNSEEN $l_p$ ATTACKS WITHOUT ADVERSARIAL TRAINING

## ABSTRACT

Deep learning has made tremendous progress in the last decades; however, it is not robust to adversarial attacks. The most effective approach is perhaps adversarial training, although it is impractical because it requires prior knowledge about the attackers and incurs high computational costs. In this paper, we propose a novel approach that can train a robust network only through standard training with clean images without awareness of the attacker's strategy. We add a specially designed network input layer, which accomplishes a randomized feature squeezing to reduce the malicious perturbation. It achieves excellent robustness against unseen $l_0, l_1, l_2$ and $l_\infty$ attacks at one time in terms of the computational cost of the attacker versus the defender through just 100/50 epochs of standard training with clean images in CIFAR-10/ImageNet. The thorough experimental results validate the high performance. Moreover, it can also defend against unlearnable examples generated by One-Pixel Shortcut which breaks down the adversarial training approach.

## 1 INTRODUCTION

Since the seminal work of Szegedy et al. (2014), the vulnerability of neural networks has been widely acknowledged by the deep learning community. A lot of solutions have been proposed to solve these problems. They can be categorized into three classes.

The first is preprocessing-based approaches that include bit-depth reduction (Xu et al., 2018), JPEG compression, total variance minimization, image quilting (Guo et al., 2018), and Defense-GAN (Samangouei et al., 2018). With this preprocessing, the hope is to reduce adversarial effect; however, it neglects the fact that the adversary can still take this operation into account and craft an effective attack through Backward Pass Differentiable Approximation (BPDA) (Athalye et al., 2018).

Secondly, perhaps the most effective method is adversarial training. In the training phase, the attack is launched through the backward gradient propagation concerning the current network state. A large volume of work falls into this class differing in ways to generate extra training samples. Madry et al. (2018) used a classical 7-step PGD attack. Other approaches are also possible, such as Mixup inference (Pang et al., 2020), feature scattering (Zhang & Wang, 2019), feature denoising (Xie et al., 2019), geometry-aware instance reweighting (Zhang et al., 2021), and channel-wise activation suppressing (Bai et al., 2021). External (Gowal et al., 2020) or generated data (Gowal et al., 2021; Rebuffi et al., 2021) are also beneficial for robustness. The inherent drawbacks are the large computation cost and the need for prior knowledge about attacks. This is certainly not realistic in practice. Also, there is a possibility of robust overfitting (Rice et al., 2020).

The last is adaptive test-time defenses. They try to purify the input iteratively as in Mao et al. (2021); Shi et al. (2021); Yoon et al. (2021) or adapt the model parameters or network structures to reverse the attack effect. For example, closed-loop control is applied in Chen et al. (2021), while a neural Ordinary Differential Equation (ODE) layer in Kang et al. (2021). Unfortunately, Croce et al. (2022) proved that most of them are not effective as claimed.

Overall, the progress is not optimistic, and marginal improvements in robust accuracy require huge computational costs while not valid for unseen attacks. So we ask a question: "can we design a novel network and loss function thereof that can drive the network to be robust on its own without awareness of adversarial attacks?" In other words, we do not intend to generate extra adversarial

samples as most other approaches do, and standard training with clean images is enough. Indeed, there should be no prior knowledge of attacks needed at all.

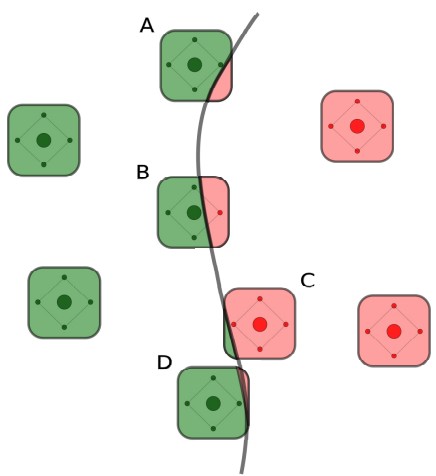

Figure 1: A conceptual illustration of feature squeezing for adversarial robustness. The rectangle around the center of a test sample represents the possible perturbation space, while only four vertices in a diamond can be allowed through feature squeezing. As a result, only B out of A, B, C, and D fails for adversarial defense, rather than all of them without feature squeezing.

This certainly poses a great challenge to the construction of networks as it is not clear even whether it is feasible. On the other hand, it appears to be possible since deep networks have a very high capacity. Unfortunately, Ilyas et al. (2019) pointed out they tend to learn discriminant features that can help correct classification, regardless of robustness. It motivates us to take the point of view from the input side. How can we make a new input layer most suitable for network robustness? Our intuition is essentially straightforward. As attacks can always walk across the class decision boundary through the malicious feature perturbations, feature squeezing might be helpful, at least reducing the space of being altered (see Figure 1 for an illustration). However, fundamentally different from the preprocessing work Xu et al. (2018), the input features are randomized squeezed with parameters learned during training as shown in Figure 2. Moreover, in the phase of the test, we simplify this layer and greatly facilitate the evaluation. The experiments of CIFAR-10 and ImageNet demonstrate this approach can promote the robustness of networks. Remarkably, although our primary motivation is adversarial defense against unseen attacks, it turns out that ours is much less influenced by the unlearnable

examples, i.e., data intentionally manipulated for unauthorized usage for training DNNs. Recently, a One-Pixel Shortcut(OPS) has been proposed in Wu et al. (2023) and could effectively degrade model accuracy even to almost an untrained counterpart even equipped with adversarial training, while ours sustains around 60%. Image Shortcut Squeezing Liu et al. (2023b) can counter OPS, however it may not deal with adversarial attacks.

With all the source codes and pre-trained models online A.2, our work has the following contributions:

- We design a special input layer that uses reciprocal and multiplication to implement our randomized feature squeezing, which is very novel. Furthermore, it could be plunged simply into networks such as WideResNet and ConvNeXt with very different structures to boost performance.

- **Our work is the only one that does not require any prior knowledge about the attacks with standard training with clean images;** while achieves great robust accuracy.

- Our approach appears to be the only one that can effectively deal with both adversarial attacks and unlearnable examples generated by the state-of-the-art OPS without any data augmentation.

## 2 RELATED WORKS

Some works add extra preprocessing steps. For example, in Yang et al. (2019), pixels are randomly dropped and reconstructed using matrix estimation. Ours is not preprocessing. Ours only adds an extra layer inside the network, and the network is trained and tested as usual without explicit image completion. Besides this, to get high robust accuracy, Yang et al. (2019) needs adversarial training, while we adopt standard training with clean images.

Another related work is certified adversarial robustness via randomized smoothing (Cohen et al., 2019). The base classifier needs Gaussian data augmentation for training, and inference is the most

likely output class of the input perturbed by isotropic Gaussian noise. Ours only uses standard training and testing, without perturbation-based training data augmentation involved at all.

Stochastic Neural Networks(SNNs) (Eustratiadis et al., 2021; Däubener & Fischer, 2022; Lee et al., 2023) achieve robustness by intentionally injecting noise into the intermediate layers of the preexisting networks, which is very different from ours. Motivated by the inherent weakness of the current network, we are trying to modify it so that adversarial defense can be achievable for both CIFAR-10 and ImagetNet for which SNNs are not available due to large image size. Unlike the usual SNN, Bart Raff et al. (2019) adopts a barrage of random transformations. However, unfortunately, its robustness is likely over-estimated as presented in Sitawarin et al. (2022).

The key-based defense such as Rusu et al. (2022); AprilPyone & Kiya (2021) is related but completely different from ours. The secure key is private and not open to the attacker, so the evaluation with a complete white-box attack is impossible. In our case, the attacker has full access to the defender's source code to launch a white-box attack.

Recently, some works have addressed the robustness from the perspective of the network's architecture. Wu et al. (2021) investigates the impact of the network width on the model robustness and proposes Width Adjusted Regularization. Similarly, Huang et al. (2021) explores architectural ingredients of adversarially robust deep neural networks thoroughly. Liu et al. (2023a) established that the higher weight sparsity benefits adversarially robust generalization via Rademacher complexity. Wang et al. (2022) proposes batch normalization removal, such that adversarial training can be improved. Singla et al. (2021) shows that using activation functions with low curvature values reduces both the standard and robust generalization gaps in adversarial training. They are in some sense similar to ours, but our motivations are fundamentally different. There is no adversarial training involved in our approach at all. We emphasize that Hou et al. (2025) reprograms the network to enhance the robustness of the baseline model; however, it only works for MNIST when there is no adversarially trained baseline model.

There are some attempts (Tramèr & Boneh, 2019; Maini et al., 2020; Croce & Hein, 2022; Laidlaw et al., 2021; Dai et al., 2022) to deal with multiple attacks simultaneously. **Among them, the only relevant works for unseen attacks are Perceptual Adversarial Training (PAT)Laidlaw et al. (2021), and adversarial training with variation regularization (AT-VR) Dai et al. (2022), but they adopt costly adversarial training, and only for CIFAR-10.** Some benchmarks (Dai et al., 2023; Kang et al., 2019) extend beyond the $l_p$ attacks.

Adversarial purification is another research line to defend against unseen attacks, but it is very slow in test. For example, in Table 14 of (Nie et al., 2022), the inference time is around (100-300)x of standard one. Also, they need pre-trained diffusion models, which are very expensive to get. Another disadvantage is that the thorough evaluation of the robustness of these methods is impossible due to very high memory consumption. Indeed, the robust accuracy is overly estimated as pointed out by Lee & Kim (2023). Moreover, adversarial purification cannot deal with OPS, as the underlying static model has very low clean accuracy.

## 3 BACKGROUND

A standard classification can be described as follows:

$$\min_{\vartheta} E_{(x,y)\sim D} \left[ L\left(x, y, \vartheta\right) \right],  \tag{1}$$

where data examples $x \in R^d$ and corresponding labels $y \in [k]$ are taken from the underlying distribution $D$, and $\vartheta \in R^p$ is the model parameters to be optimized with respect to an appropriate function $L$, for instance cross-entropy loss. When $x \in R^d$ can be maliciously manipulated within a set of allowed perturbations $S \subseteq R^d$, which is usually chosen as a $l_p$-ball ($p \in \{0, 1, 2, \infty\}$) of radius $\epsilon$ around $x$, Equation 1 should be modified as:

$$\min_{\vartheta} E_{(x,y)\sim D} \left[ \max_{\delta \in S} L\left(x + \delta, y, \vartheta\right) \right].  \tag{2}$$

An adversary implements the inner maximization via various white-box or black-box attack algorithms, for example, APGD-ce (Croce & Hein, 2020) or Square Attack (Andriushchenko et al., 2020).

The basic multi-step projected gradient descent (PGD) is

$$x^{t+1} = \Pi_{x+S}\left(x^t + \alpha \operatorname{sgn}\left(\nabla_x L\left(x, y, \vartheta\right)\right)\right), \tag{3}$$

where $\alpha$ denotes a step size and $\Pi$ is a projection operator. In essence, it uses the current gradient to update $x^t$, such that a better adversarial sample $x^{t+1}$ can be obtained. Some heuristics can be used to get better gradient estimation in Croce & Hein (2020). On the other hand, outer minimization is the goal of a defender.

Adversarial training is the most effective approach to achieve this outer minimization via augmenting the training data with crafted samples. In fact, all current approaches, including test-time adaptive defense as it needs a base classifier, aim to learn the parameters of a pre-existing model to improve the robustness. In this paper, we try to increase the robustness through a specially designed input layer such that standard training with clean images can be adopted.

# 4 METHOD

## 4.1 INPUT LAYER

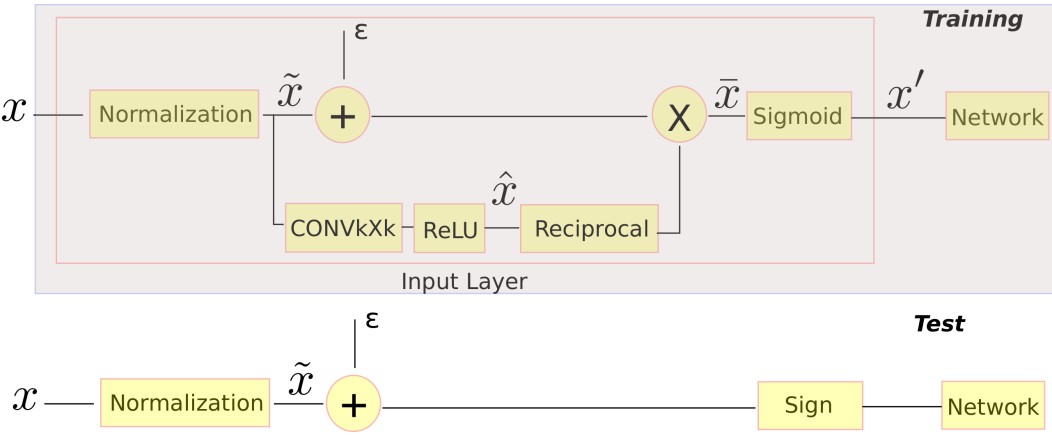

Figure 2: The shaded and non-shaded areas show the training and test framework respectively. In training, our specially designed input layer is inside the red rectangle. The input image $x$ is first normalized, then undergoes two paths. On one path, independent Gaussian noise $\epsilon$ is added, and the other path includes $k \times k$ convolution and ReLU followed by element-wise reciprocal. Finally, these two terms are combined through element-wise multiplication and the result feeds to the Sigmoid. The final $x'$ will be used as inputs to the classification network, the same as other training approaches. End-to-end training scheme is adopted to learn the parameters of $k \times k$ convolution. In test, the conv $k \times k$ path is removed wholly, and Sigmoid is replaced with Sign defined in Equation 5.

As we stated earlier, the goal of input layer is to squeeze the input feature in a random and controlled way. The whole procedure is depicted in Figure 2.

It consists of the following steps:

1. The input $x$ with $r, g, b$ channel will be normalized to a variable with a mean 0 and a standard deviation 1, through $\tilde{x} = \frac{x - mean}{std}$ in the input layer, where $mean$ and $std$ are mean and standard deviation of training set. Then, it goes through top and bottom paths.

2. In the top path, each element of $\tilde{x}$ is corrupted independently by additive Gaussian noise $\varepsilon$, where $\varepsilon \sim N\left(0, \sigma^2\right)$.

3. In the bottom path, $\tilde{x}$ goes through a $k \times k$ 2D convolution and ReLU, and we get $\hat{x}$ with three channels, and then its element-wise reciprocal $\frac{1}{\hat{x}+\gamma}$, where $\gamma$ is a small constant in order to make the denominator always positive, which is $1 \times 10^{-5}$ in this paper.

4. The top and bottom paths are combined by element-wise multiplication, $\bar{x} = (\tilde{x} + \varepsilon) \times \frac{1}{\hat{x}+\gamma}$.

5. The final output $x'$ is a Sigmoid of the $\bar{x}$, i.e., $x' = \frac{1}{1+\exp(-\bar{x})}$.

So essentially,

$$x' = \frac{1}{1 + \exp\left(-\frac{(\tilde{x}+\varepsilon)}{\hat{x}+\gamma}\right)}. \tag{4}$$

This formula can be interpreted this way. $\tilde{x} + \varepsilon$ is a polluted version of the input image, and $\frac{1}{\hat{x}+\gamma}$ tries to modulate the image based on the $\hat{x}$, named as sampling matrix having the same size as input $x$.

The key motivation is that if we enforce $\hat{x}$ to be very small through some loss function, $\left|\frac{(\tilde{x}+\varepsilon)}{\hat{x}+\gamma}\right|$ will become big and the response of Sigmoid will be on the saturated region, i.e., most elements of $x'$ will be either 0 or 1. In other words, the input feature will be squeezed in a random manner where the parameters of sampling matrix $\hat{x}$ are learned on the end-to-end training.

Accordingly, $\varepsilon$ plays the role of mimicking the attack that the adversary may launch. The appropriate value of $\sigma$ should be chosen as the big one will degrade the clean accuracy while the network can not gain much robustness for the small one.

Based on our analysis above, one may raise a big concern regarding the obfuscated gradients Athalye et al. (2018) which may be incurred by reciprocal and Sigmoid operator in robustness evaluation. On one hand, $\hat{x}$ is very small, so the gradient of reciprocal $\frac{1}{\hat{x}+\gamma}$ will be very big. On the other hand, $\bar{x} = (\tilde{x} + \varepsilon) \times \frac{1}{\hat{x}+\gamma}$ will reside on the saturated domain of Sigmoid, i.e., the gradients of $x'$ with respect to $\bar{x}$ will be very small. Actually, this might also cause some trouble in training, as we need to learn the parameters of conv $k \times k$ for sampling matrix $\hat{x}$, although they might be canceled out by each other to some extent, as they are on the same path in backward pass gradient propagation.

To resolve this, in training we adopt the BPDA-like optimization procedure. Namely, for the forward pass, we evaluate the reciprocal and Sigmoid as usual, however, in the backward pass, the gradient of the reciprocal is set to be -1, and 1 for Sigmoid. While in test, because Sigmoid often goes to two extreme values 0 and 1, dependent of the sign of $\tilde{x} + \varepsilon$, we just remove the bottom path wholly, and replace the Sigmoid with Sign which is defined as:

$$Sign(x) = \begin{cases} 0, & \text{if } x < 0, \\ 0.5, & \text{if } x = 0, \\ 1, & \text{if } x > 0. \end{cases} \tag{5}$$

This will greatly simplify our robust evaluation. Of course, it is also possible to train directly in the test framework. The detailed analysis is deferred to Section 5.3. In essence, the proposed training architecture introduces a refined optimization protocol for the target test framework. Rather than direct parameter tuning, this detour approach strategically incorporates auxiliary components—conv$k \times k$ layers, relu, reciprocal, and sigmoid—coupled with a customized loss function discussed in the next section. These transient modules are exclusively employed during the training and subsequently discarded during inference, yielding a streamlined test framework characterized by random noise injection and Sign.

## 4.2 Loss Function

As mentioned earlier, we have to design a loss function to implement our motivation to make the sampling matrix $\hat{x}$ small. For each $\hat{x}$, we get $S$, the average of all the elements of $\hat{x}$ that are greater than some threshold $\beta$. Formally,

$$S = \frac{\sum\limits_{i \in T} \hat{x}_i}{\#T}, \quad where \ T = \{i | \hat{x}_i > \beta\}. \tag{6}$$

A small $\beta$ means $\hat{x}$ will become sparse. The final loss function is:

$$L = \alpha \times L_{ce} + S, \tag{7}$$

where $S$ is the sparse loss and $L_{ce}$ is the cross-entropy loss with weight $\alpha$. When $\alpha$ becomes large, the loss function falls back to standard cross-entropy. In summary, there are only four parameters, $\sigma$ of noise, the size of convolution kernel $k$, weight $\alpha$, and threshold $\beta = 0.2$ in this paper.

## 5 EXPERIMENTS

To verify the effectiveness of our approach, we conducted the experiments on both CIFAR-10 and ImageNet.

For CIFAR-10, we choose the wide residual network WideResNet-28-10 (Zagoruyko & Komodakis, 2016) as the base one, where we add our specially designed input layer as described in Section 4 with $\sigma = 0.65$, $\alpha = 0.1$, and conv $5 \times 5$. The initial learning rate of 0.1 is scheduled to drop at 30, 60, and 80 out of 100 epochs in total with a decay factor of 0.2. The weight decay factor is set to $5 \times 10^{-4}$, and the batch size is 200. To emphasize again, we only perform standard training through just 100 epochs. **Reemphasize that there is no work for unseen attacks with standard training, and if adversarial training is allowed, Laidlaw et al. (2021) and Dai et al. (2022) are the only two to this end with training costs 16 and 62 times as of ours with the same WideResNet-28-10, as shown in Table 1.**

ImageNet is the most challenging dataset for adversarial defense, and **there is no work dealing with unseen attacks even with adversarial training.** In this paper, ImageNet only refers to ImageNet-1k without explicit clarification, and robustness is only evaluated on the ImageNet validation set. For simplicity, we choose the architecture of ConvNeXt-T + ConvStem in Singh et al. (2023) with $\sigma = 1.4$, $\alpha = 0.5$, and conv $7 \times 7$. Our training scheme is very simple. All parameters are randomly initialized, followed by standard training for 50 epochs with heavy augmentations without CutMix (Yun et al., 2019) and MixUp (Zhang et al., 2018), as these will undermine the viability of our sampling matrix. While for the same ConvNeXt-T + ConvStem in Singh et al. (2023), although ConvStem is randomly initialized, the ConvNeXt-T part is from a strong pre-trained model which usually takes about 300 epochs. Thus the whole network needs extra standard training for 100 epochs to get good clean accuracy, followed by 300 epochs of adversarial training with 2-step APGD. So the total cost is up to $300 + 100 + 300 \times [2 \, (\text{APGD steps}) + 1 \, (\text{weights update})] = 1300$, which is around $1300/50 = 26$ times bigger than ours.

Table 1: Clean and training cost comparison. For CIFAR10, the cost is defined as: #Epochs $\times$ [#PGD $+ 1$ (weights update)] with respect to ours, which is denoted by 1. For ImageNet, please refer to the above main text. Since ours is random, we report mean and standard deviation for five runs.

| Defense | Clean | #Epochs | #PGD | #Cost |
|---|---|---|---|---|
| CIFAR-10 | | | | |
| Ours | 80.23±0.30 | 100 | 0 | 1 |
| PATLaidlaw et al. (2021) | 82.40 | 100 | 15 | 16 |
| AT-VRDai et al. (2022) | 72.73 | 200 | 30 | 62 |
| ImageNet | | | | |
| Ours | 67.60±0.55 | 50 | 0 | 1 |
| Singh et al. (2023) | 72.74 | 400+300 | 2 | 26 |

As expected, our specially designed input layer changes the input $x$ into $x'$ that are extremely squeezed. On one hand, it poses a great challenge to the network; while on the other hand, it improves the robustness. Some of the example feature maps in our input layers are listed in Figure 3. Notably, thanks to the great capacity of deep network, our defense achieves reasonably good and stable clean accuracy on total images, i.e., 10k for CIFAR-10, and 5K for ImageNet; and due to resource constraints, we will demonstrate the robustness performance on only 1K images unless explicitly specified, which is enough to make a fair comparison, against $l_0, l_1, l_2$ and $l_\infty$ attacks in the follow sections with $l_1$-$\epsilon$=12, $l_2$-$\epsilon$=1, $l_\infty$-$\epsilon$=8/255 for CIFAR-10; and $l_1$-$\epsilon$=75, $l_2$-$\epsilon$=2, $l_\infty$-$\epsilon$=4/255 for ImageNet.

**The attack for a determined network only accepts correctly classified clean images and stops further operation once the network gets fooled. Since ours is random, we run out of the maximum**

**allowed number of iterations for all input samples to ensure that the generated adversarial samples have a high probability of fooling the network. This principle goes throughout all experiments for scientific rigor.**

## 5.1 BLACK-BOX ATTACKS

For $l_0$ we use Pomponi et al. (2022), which is based on rearranging the pixels inside a random selected patch without limits of the number of perturbed pixels, i.e., $l_0$ norm. The configuration for CIFAR-10 is 25 restarts with 10 max iterations per restart with patch dimension of 3, so the total budget is 250 iterations; while for ImageNet, 100 restarts with 50 max iterations per restart with the same dimension 3, accordingly 5000 in total. For CIFAR-10, the average iterations of competitors are only half of ours while the accuracies drop to random guess with $l_0$ less than ours. please refer to Table 6 for more details.

For $l_1$ we use Square Andriushchenko et al. (2020), which is commonly adopted in adversarial defense evaluation. We use the default iterations 5000. For $l_2, l_\infty$, we use both Square Andriushchenko et al. (2020) and SignHunter Al-Dujaili & O'Reilly (2020), which is a divide-and-conquer, adaptive, memory-efficient algorithm.

As in Table 2, $l_0, l_1, l_2,$ and $l_\infty$ black-box attacks are almost impotent to ours, and in general our robust accuracy significantly outperforms others by a large margin except in $l_2$ for ImageNet. Our high performance comes from the very effective adversarial perturbation simulation during training operated by random noise, which also plays a role in misleading the attack in the test.

Table 2: Robustness comparison against $l_0, l_1, l_2,$ and $l_\infty$ black-box attacks. The iterations column shows the average number of iterations by the attack. Two columns in $l_2$ and $l_\infty$ are respecively for Square and SignHunter (in italic). Adv-Trained refers to Singh et al. (2023).

| Defense | Robust | | | | | | Iterations | | | | | |
|---|---|---|---|---|---|---|---|---|---|---|---|---|
| CIFAR-10 | $l_0$ | $l_1$ | $l_2$ | | $l_\infty$ | | $l_0$ | $l_1$ | $l_2$ | | $l_\infty$ | |
| Ours | 61.10 | 79.00 | 77.60 | *78.70* | 76.70 | *77.80* | 250 | 5000 | 5000 | *5000* | 5000 | *5000* |
| PAT | 12.00 | 52.90 | 62.20 | *61.20* | 46.00 | *42.10* | 122 | 3480 | 4026 | *3939* | 3230 | *2961* |
| AT-VR | 11.40 | 27.70 | 53.20 | *60.80* | 53.10 | *51.50* | 109 | 2390 | 3915 | *4263* | 3807 | *3657* |
| ImageNet | $l_0$ | $l_1$ | $l_2$ | | $l_\infty$ | | $l_0$ | $l_1$ | $l_2$ | | $l_\infty$ | |
| Ours | 67.70 | 67.00 | 66.90 | *68.40* | 65.40 | *67.40* | 5000 | 5000 | 5000 | *10000* | 5000 | *10000* |
| Adv-Trained | 33.40 | 50.70 | 69.00 | *71.00* | 64.30 | *64.30* | 3227 | 3668 | 4791 | *9752* | 4521 | *9001* |

## 5.2 WHITE-BOX ATTACKS

Since Sign is non-differentiable, the backward pass differentiation should be approximated with some function to evaluate the robust accuracy on white-box attacks. We have tried different options for 1K samples: identity; $\frac{\text{Softsign}(x)+1}{2}$, where $\text{Softsign}(x) = \frac{x}{1+|x|}$; and Sigmoid($ax$), $a \in \{1,3,5,7,9\}$. Rusu et al. (2022) also used Sign, and only tested Softsign($x$) and Sigmoid($x$). Of course, more options can convince us more of the robustness of our defense. We have tested against attacks of APGD-$l_2$ and APGD-$l_\infty$, but their robust accuracies are higher than the plain version. Indeed, the similar observation is also reported in Lee & Kim (2023). The APGD-$l_1$ is adopted since it is stronger than Sparse $l_1$ Tramèr & Boneh (2019). The results are shown in Table 8. According to the worst accuracies among all BPDA in Table 8, to ensure the legitimate robust evaluation in the following sections, for CIFAR-10, we select $\frac{\text{Softsign}(x)+1}{2}$, for both APGD-$l_1$ and PGD-$l_2$, Sigmoid($3x$) for PGD-$l_\infty$; while for ImageNet, $x$ for APGD-$l_1$, $\frac{\text{Softsign}(x)+1}{2}$ for PGD-$l_2$, and Sigmoid($3x$) for $l_\infty$. Both APGD-$l_1$ (5 restarts; while 1 for EOT) and PGD (1 restart) have 100 steps in this paper.

Based on the comparison in Table 3, ours converges on EOT-20. Note that EOT incurs a high computational cost, so it is apparent in the inferior status to compare the robustness with others, but a good performance is held, especially for $l_1$ on ImageNet. For more comparisons with DDN Rony et al. (2019), C&W Carlini & Wagner (2017), and Spatial Transform Xiao et al. (2018) attacks, please refer to A.6.

Table 3: Robustness against $l_1, l_2,$ and $l_\infty$ white-box attacks on total images except the rows of EOT only for 1k images.

| Defense | Clean | APGD-$l_1$ | PGD-$l_2$ | PGD-$l_\infty$ |
|---|---|---|---|---|
| CIFAR-10 | BPDA | $\frac{\text{Softsign}(x)+1}{2}$ | $\frac{\text{Softsign}(x)+1}{2}$ | Sigmoid($3x$) |
| Ours | 80.23 | 57.16 | 37.97 | 42.38 |
| Ours-EOT20 | 81.02 | 38.50 | 33.90 | 34.00 |
| Ours-EOT50 | 81.02 | 38.00 | 34.50 | 32.50 |
| PATLaidlaw et al. (2021) | 82.40 | 33.22 | 40.96 | 36.38 |
| AT-VRDai et al. (2022) | 72.73 | 9.06 | 31.21 | 52.73 |
| ImageNet | BPDA | $x$ | $\frac{\text{Softsign}(x)+1}{2}$ | Sigmoid($3x$) |
| Ours | 67.60 | 57.06 | 40.46 | 24.18 |
| Ours-EOT20 | 68.80 | 46.90 | 37.90 | 18.80 |
| Ours-EOT50 | 68.80 | 46.60 | 38.10 | 17.40 |
| Singh et al. (2023) | 72.74 | 30.63 | 53.56 | 53.28 |

### 5.2.1 ONE-PIXEL SHORTCUT

Although our approach is motivated for adversarial defense, it turns out ours is much less impacted by OPS without any data augmentation. Following the OPS Wu et al. (2023), we also choose ResNet-18 and all training settings are exactly the same as WideResNet-28-10 except for $\tau = 0.3$ and conv $3 \times 3$. Ours exceeds others by 40+ in Table 7. Again, it is due to the random featured squeezing. Since we transform all pixels to 1 or 0, the pixel chosen by the OPS can not stand out from its neighbors.

### 5.3 ABLATION STUDIES

Training in the configuration of the test can be regarded as special case of our detour training framework where the parameters of all the conv $k \times k$ for sampling matrix $\hat{x}$ are manually set to be zero, then the reciprocal will become very big, and the Sigmoid is approximately equal to Sign. Table 4 shows that this training scheme can achieves robust accuracy since it also implements random feature squeezing. However, it is always inferior to normal ones, especially for CIFAR-10; while performance gap is small on ImageNet. This could be due to the following reasons. With normal training, at the early stages, since the sparse loss $S$ in Equation 7 is relatively big, the feeds to the network, $x'$, is not highly squeezed to be 0 or 1 at the early stages, which seems to benefit robustness. For ImageNet, compared with CIFAR-10, due to the massive size of the datasets and the unique ConvNeXt-T+ConvStem structure, $S$ drops quickly and $x'$ goes to the extreme values much faster, thus only having marginal improvement. Please refer to Section A.8 for more analysis.

Table 4: Comparison between the normal training(N.T.) and training with test(T.T.) framework. The Clean is for total images, while the robust accuracy is on 1k.

| Defense | Clean | APGD-$l_1$ | PGD-$l_2$ | PGD-$l_\infty$ |
|---|---|---|---|---|
| CIFAR-10 | | | | |
| N.T. | 80.23±0.30 | 65.00 | 39.90 | 41.70 |
| T.T. | 77.92±0.18 | 58.10 | 31.50 | 31.80 |
| ImageNet | | | | |
| N.T. | 67.60±0.55 | 65.80 | 42.40 | 25.40 |
| T.T. | 66.90±0.31 | 64.70 | 40.80 | 23.90 |

Another possible concern may be related with transfer attack, i.e., using the training framework as the targeted net to generate adversarial examples. This is not effective as shown in Section A.7, since training network contains the sigmoid and reciprocal plus small conv $k \times k$, which is driven by our sparse loss term. This will cause some problems in gradient backpropagation, even though BPDA is adopted as done in training.

## 6    ROBUST ACCURACY FOR DETERMINISTIC MODEL

Table 5: Accuracy for APGD attack for the 1k images from CIFAR-10 in both training and test sets (in bold) with different $N$. $\mathcal{S}$ stands for Sigmoid.

| $N$ | | Clean | APGD-$l_1$ | APGD-$l_2$ | APGD-$l_\infty$ |
|---|---|---|---|---|---|
| CIFAR-10 | | BPDA | $\mathcal{S}(9x)$ | $\mathcal{S}(19x)$ | $\mathcal{S}(19x)$ |
| 5 | | 99.80 **83.80** | 10.20 **6.90** | 4.60 **2.80** | 26.00 **20.30** |
| 20 | | 99.90 **84.90** | 22.40 **17.50** | 15.00 **12.20** | 30.80 **24.80** |
| 30 | | 99.90 **85.60** | 24.20 **19.20** | 20.30 **16.40** | 32.50 **25.80** |
| ImageNet | | BPDA | $\mathcal{S}(25x)$ | $\mathcal{S}(25x)$ | $\mathcal{S}(25x)$ |
| 5 | | 84.70 **70.20** | 7.30 **6.70** | 0.40 **0.10** | 3.50 **3.10** |
| 20 | | 85.40 **70.30** | 37.20 **27.40** | 7.80 **6.70** | 5.90 **5.30** |
| 30 | | 85.50 **70.10** | 45.60 **32.70** | 13.70 **10.90** | 7.00 **6.90** |

Although we have demonstrated excellent experimental performance, one may still wonder why this is possible, especially for those who are uncomfortable with randomness. Here, we remove randomness and transform the test framework with random noise into a deterministic one. To our knowledge, no existing work comprising a random component has been evaluated with that component fixed. More specifically, we feed the $N$ copies of the same test image to the test framework, each with a different but fixed seed of noise, and then the average logits of $N$ outputs are used to get the final classification. It might be possible that our training scheme implements implicit adversarial training due to the added random noise; however, it is unclear how it relates to test robust accuracy. It appears that feature squeezing is beneficial in this regard, as it reduces the adversarial perturbation space of the test sample, thereby diminishing the negative impact of the out-of-distribution effect. Since it is a deterministic model, we can safely use APGD for $l_{1,2,\infty}$ attacks, and BPDA is also quite different from previous ones since we find that these BPDA can support stronger attacks. For more details, please refer to Table 11 in the Appendix.

Interestingly, as expected, non-trivial training robust accuracy is achieved through the implicit adversarial training with our specially designed input layer with the random noise, and thanks to the feature squeezing, test robust accuracy also keeps up, with maximum discrepancy less than 10 for most cases shown in Table 5. For ImageNet, ours achieves a higher $l_1$ accuracy than Singh et al. (2023), a remarkable evidence that standard training can outperform adversarial one.

Now we give more thorough analysis. In ImageNet, each image has dimensions of 224×224×3 (height×width×channels). Assuming an 8-bit depth per channel, the total number of possible distinct images in the input space is $2^{224 \times 224 \times 3 \times 8}$. However, our method constrains this space to $2^{224 \times 224 \times 3}$, achieving an exponential compression ratio of $2^{224 \times 224 \times 3 \times 7}$. This dramatic dimensionality reduction inherently limits the adversarial perturbation space while preserving essential image semantics to some extent, and indeed, ours achieves a good clean accuracy. Moreover, the random component in our design enables exploration of this compressed space, which leads to enhanced model robustness. This improvement manifests in training data and generalizes to test sets. Notably, this robustness mechanism operates independently of gradient obfuscation techniques, instead deriving from the intrinsic properties of our compressed representation space. The loss landscapes in Section A.10 also verify this.

## 7    SUMMARY

In this paper, we proposed an efficient and effective method for unseen attacks only through standard training. To our knowledge, this is the only paper that falls within this category.

There are several possible future research directions. Firstly, the clean accuracy needs to be improved. Secondly, the efficient noise injection scheme should be investigated in order to improve $l_2$ and $l_\infty$ robust accuracy. Thirdly, the strong theoretical robustness guarantee is preferred.

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

# A APPENDIX

## A.1 FEATURE MAP

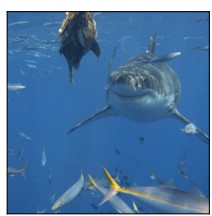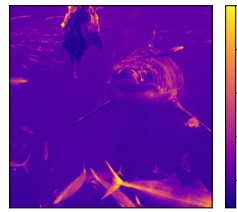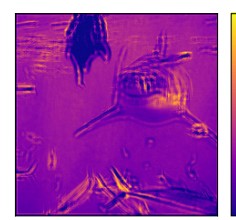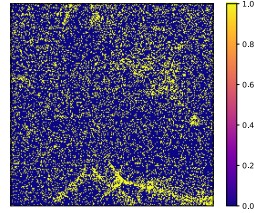

Figure 3: From the left to right are the great-white-shark $x$; the red channel; and the corresponding sampling matrix $\hat{x}$ and the final output $x'$ where the continuous patterns are highly squeezed into two extreme values, 0 and 1, due to very small $\hat{x}$. Blue and green channels share a similar situation.

## A.2 SOURCE CODES AND PRE-TRAINED MODELS

1. CIFAR-10

   `https://rfsq.obs.cn-north-4.myhuaweicloud.com/cifar.zip`

2. ImageNet

   `https://rfsq.obs.cn-north-4.myhuaweicloud.com/imagenet.zip`

## A.3 $l_0$ ATTACK

Table 6: $l_0$ Robustness comparison. There is no specific constrains for $l_0$, and last column shows the average of $l_0$ of perturbed samples; while the iterations column shows the average number of iterations by the attack.

| Defense | Clean | Robust | Iterations | $l_0$ |
|---|---|---|---|---|
| CIFAR-10 | | $l_0$ | $l_0$ | |
| Ours | 81.02 | 61.10 | 250 | 122 |
| PATLaidlaw et al. (2021) | 82.30 | 12.00 | 122 | 101 |
| AT-VRDai et al. (2022) | 73.00 | 11.40 | 109 | 92 |
| ImageNet | | $l_0$ | $l_0$ | |
| Ours | 68.80 | 67.70 | 5000 | 1468 |
| Singh et al. (2023) | 73.40 | 33.40 | 3227 | 1706 |

## A.4 OPS

Table 7: Performance under One-Pixel Shortcut on ResNet-18 for different training strategies. The first two rows are excerpted from Wu et al. (2023). $l_\infty$ AT stands for adversarial training with $l_\infty$=8/255.

| Training Strategy | Clean | OPS |
|---|---|---|
| Standard | 94.01 | 15.56 |
| $l_\infty$ AT | 82.72 | 11.08 |
| Ours | 82.03 | 56.25 |

## A.5 BPDA FOR RANDOM MODEL

Table 8: Comparison between the different BPDA. We choose the worst ones (in bold ) as BPDA for tests of our defense. $\mathcal{S}$ stands for Sigmoid. The attacks of APGD including DLR loss are much weaker than PGD ones. Please note that without an explicit statement, we adopt cross-entropy loss in this paper.

| DataSet | Attack | $x$ | $\frac{\text{softsign}(x)+1}{2}$ | $\mathcal{S}(x)$ | $\mathcal{S}(3x)$ | $\mathcal{S}(5x)$ | $\mathcal{S}(7x)$ | $\mathcal{S}(9x)$ |
|---|---|---|---|---|---|---|---|---|
| | APGD-$l_1$ | 63.50 | **57.00** | 57.60 | 60.00 | 65.00 | 66.50 | 69.00 |
| CIFAR-10 | PGD-$l_2$ | 43.10 | **36.40** | 39.40 | 37.60 | 39.90 | 38.00 | 38.90 |
| | APGD-$l_2$ | 51.50 | 52.90 | 52.20 | 51.80 | 51.20 | 52.20 | 49.70 |
| | APGD$^{\text{DLR}}$-$l_2$ | 56.20 | 58.00 | 55.70 | 56.20 | 57.80 | 55.50 | 55.30 |
| | PGD-$l_\infty$ | 41.90 | 42.50 | 41.20 | **41.10** | 41.70 | 42.60 | 42.10 |
| | APGD-$l_\infty$ | 53.70 | 55.40 | 54.40 | 53.40 | 54.70 | 54.60 | 57.40 |
| | APGD$^{\text{DLR}}$-$l_\infty$ | 60.20 | 59.80 | 59.60 | 59.00 | 58.40 | 59.50 | 57.80 |
| | APGD-$l_1$ | **58.20** | 61.90 | 59.20 | 64.10 | 65.80 | 66.50 | 67.10 |
| ImageNet | PGD-$l_2$ | 41.70 | **38.90** | 40.20 | 39.40 | 42.40 | 43.60 | 44.60 |
| | APGD-$l_2$ | 49.20 | 50.60 | 47.90 | 51.80 | 55.10 | 55.50 | 55.60 |
| | APGD$^{\text{DLR}}$-$l_2$ | 50.40 | 53.80 | 50.90 | 55.20 | 58.20 | 58.90 | 60.50 |
| | PGD-$l_\infty$ | 23.70 | 24.70 | 24.50 | **23.50** | 25.40 | 25.30 | 23.60 |
| | APGD-$l_\infty$ | 34.10 | 33.10 | 33.40 | 34.80 | 34.80 | 32.70 | 35.50 |
| | APGD$^{\text{DLR}}$-$l_\infty$ | 37.30 | 40.40 | 39.30 | 37.20 | 39.10 | 39.40 | 38.20 |

## A.6 MORE COMPARISONS

Table 9: Robust accuracy against DDN, C&W and Spatial Transform attacks. It is interesting to note that all other approaches fail against DDN attacks, while ours sustain.

| Defense | Clean | DDN | C&W | Spatial Transform |
|---|---|---|---|---|
| CIFAR-10 | | | | |
| Ours | 81.02 | 58.30 | 68.00 | 17.50 |
| PAT Laidlaw et al. (2021) | 82.30 | 0.00 | 64.60 | 4.80 |
| AT-VR Dai et al. (2022) | 73.00 | 0.10 | 46.50 | 10.30 |
| ImageNet | | | | |
| Ours | 68.80 | 51.50 | 66.80 | 44.50 |
| Singh et al. (2023) | 73.40 | 2.90 | 67.10 | 1.20 |

### A.7 TRANSFER ATTACK

The transfer attack is launched using the training framework, which is weaker than a direct attack on the test framework, except for its closeness with PGD-$l_\infty$.

Table 10: Robustness against EOT-20 transfer attack.

| Defense | APGD-$l_1$ | PGD-$l_2$ | PGD-$l_\infty$ |
|---|---|---|---|
| CIFAR-10 | | | |
| Transfer-EOT20 | 55.70 | 51.90 | 34.30 |
| Normal-EOT20 | 38.50 | 33.90 | 34.00 |
| ImageNet | | | |
| Transfer-EOT20 | 52.00 | 47.50 | 18.70 |
| Normal-EOT20 | 46.90 | 37.90 | 18.80 |

### A.8 DETOUR TRAINING

The key distinction between the standard test framework training and detour learning lies in the initialization strategy and early-phase optimization dynamics. In training with the test framework where parameters are randomly initialized, $x'$ saturates to binary values (0/1) at the start. This premature squeezing limits the model's exploration capacity, potentially trapping it in suboptimal local minima. By contrast, detour learning introduces a warm-up phase where $x'$ is not highly squeezed to be 0 or 1, allowing parameters to discover better initialization regions. After that point, $x'$ is highly squeezed to 0/1, so it does the training with the test framework to the same effect.

The following Figure 4 shows that although the cross-entropy loss shows similar trends, the sparse loss is very different, slowly for CIFAR-10 drops while quick for ImagetNet. This sparse loss evolution indicates that detour training for ImageNet is closer to direct training possibly due to massive image size and ConvNeXt-T+ConvStem structure, which leads to only marginal improvement.

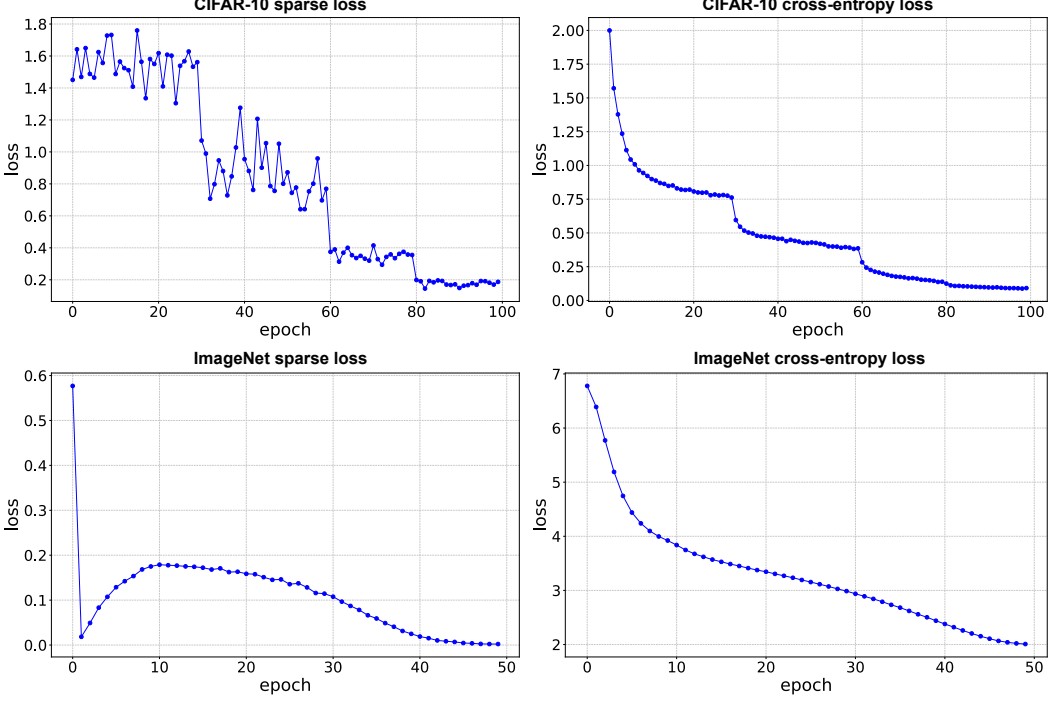

Figure 4: From the left to right are the sparse loss and cross-entropy loss.

## A.9 BPDA FOR DETERMINISTIC MODEL

Table 11: Comparison between the different BPDA on 1k images for deterministic model with $N = 5$. We choose the worst ones (in bold ) as BPDA for tests of our defense. $\mathcal{S}$ stands for Sigmoid.

| DataSet | Attack | $x$ | $\frac{\text{softsign}(x)+1}{2}$ | $\mathcal{S}(x)$ | $\mathcal{S}(3x)$ | $\mathcal{S}(5x)$ | $\mathcal{S}(7x)$ |
|---|---|---|---|---|---|---|---|
| | APGD-$l_1$ | 32.70 | 10.70 | 22.20 | 11.00 | 9.10 | 7.70 |
| CIFAR-10 | APGD-$l_2$ | 34.60 | 14.00 | 28.50 | 17.20 | 9.80 | 6.60 |
| | APGD-$l_\infty$ | 27.60 | 23.70 | 26.70 | 24.10 | 22.60 | 22.20 |
| | | $\mathcal{S}(9x)$ | $\mathcal{S}(13x)$ | $\mathcal{S}(17x)$ | $\mathcal{S}(19x)$ | $\mathcal{S}(21x)$ | |
| | APGD-$l_1$ | **6.90** | 7.20 | 7.30 | 7.40 | 8.20 | |
| CIFAR-10 | APGD-$l_2$ | 5.00 | 3.30 | 2.90 | **2.80** | 3.40 | |
| | APGD-$l_\infty$ | 21.50 | 20.70 | 20.80 | **20.30** | 20.90 | |
| | | $x$ | $\frac{\text{softsign}(x)+1}{2}$ | $\mathcal{S}(x)$ | $\mathcal{S}(3x)$ | $\mathcal{S}(5x)$ | $\mathcal{S}(7x)$ |
| | APGD-$l_1$ | 35.60 | 18.00 | 27.60 | 18.70 | 15.30 | 12.10 |
| ImagNet | APGD-$l_2$ | 32.60 | 11.30 | 26.10 | 14.00 | 8.60 | 5.90 |
| | APGD-$l_\infty$ | 10.10 | 5.40 | 8.20 | 5.50 | 4.60 | 4.00 |
| | | $\mathcal{S}(9x)$ | $\mathcal{S}(15x)$ | $\mathcal{S}(20x)$ | $\mathcal{S}(25x)$ | $\mathcal{S}(30x)$ | |
| | APGD-$l_1$ | 10.40 | 7.40 | 6.90 | **6.70** | 7.30 | |
| ImagNet | APGD-$l_2$ | 3.70 | 0.60 | 0.30 | **0.10** | 0.10 | |
| | APGD-$l_\infty$ | 3.90 | 3.40 | 3.20 | **3.10** | 3.10 | |

## A.10 LOSS LANDSCAPES

The loss landscapes in Figure 5 generated using the code adapted from Eustratiadis et al. (2022) show that although a random version of our network exhibits a rough surface, it becomes smoother as the iterations of EOT increase. The determined versions are smooth. It suggests that EOT+BPDA is enough to give a robust evaluation without the risk of overestimation.

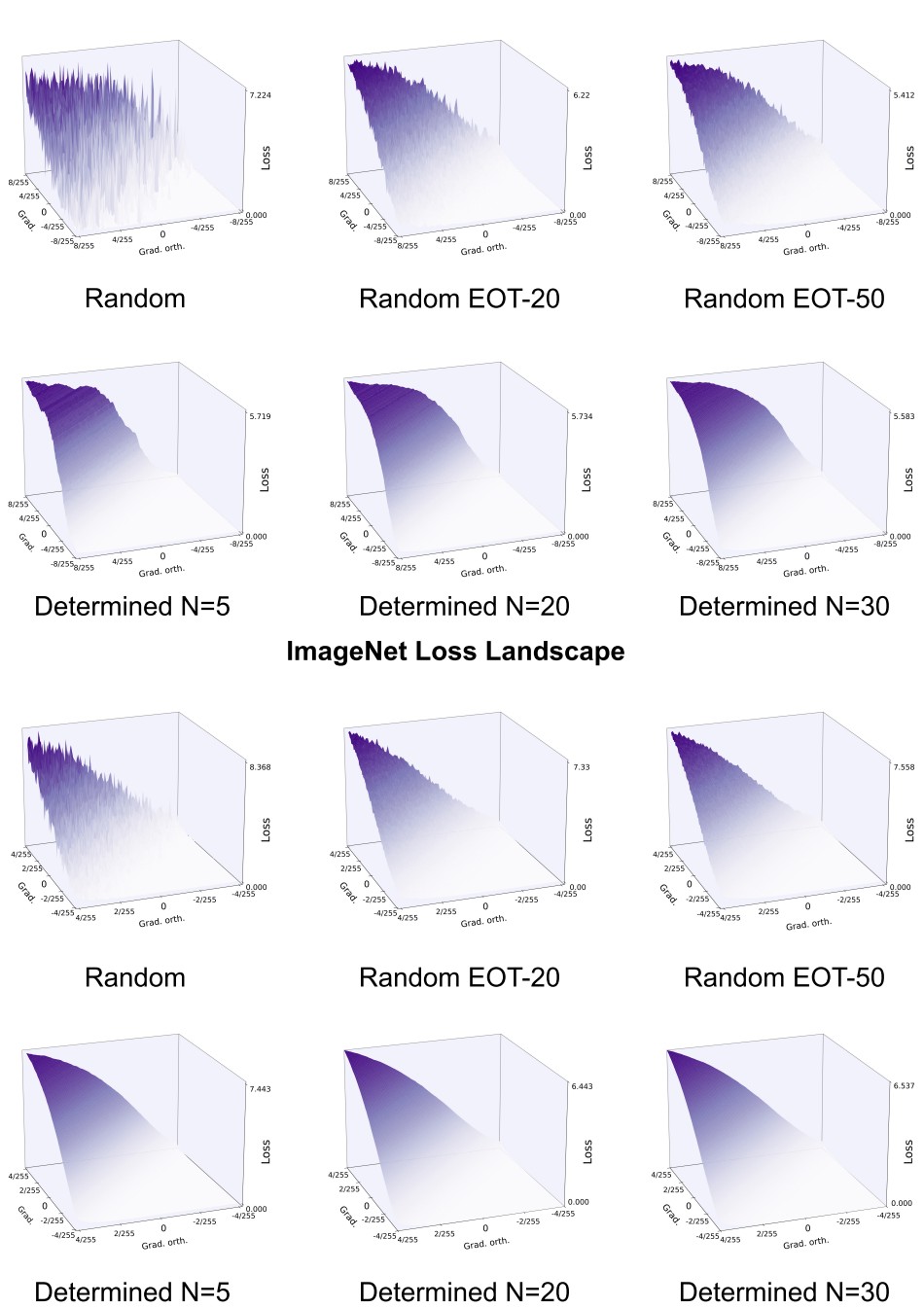

Figure 5: The loss landscapes of our defense.

