# OpenReview forum: "Randomized Feature Squeezing against  Unseen  $ {l_p} $ Attacks without Adversarial Training"
_ICLR.cc/2026/Conference — Submitted to ICLR 2026_

### Official Review · Reviewer_NjGZ · 2025-10-16

**Soundness:** 1
**Presentation:** 2
**Contribution:** 2
**Rating:** 2
**Confidence:** 5

**Summary:**

For more effective and efficient defense against adversarial attacks, this paper introduces a novel approach that can train a robust network only through standard training with clean images without awareness of the attacker’s strategy. The proposed method achieves excellent robustness against unseen attacks at one time in terms of the computational cost of the attacker versus the defender through just 100/50 epochs of standard training with clean images in CIFAR-10/ImageNet.

**Strengths:**

1. The paper is easy to understand.
2. Experiments have been conducted on both CIFAR-10 and ImageNet.

**Weaknesses:**

1. The writing quality needs improvement.
2. The description in Section 4.1 is too subjective, lacks theoretical or comprehensive empirical support, and shows little innovation.
3. The font size in Table 2 is too large, causing the numbers to merge together, which makes it difficult to read.
4. The results in Table 2 are confusing. All accuracy are shown as xx.x0%. It seems that the evaluation is performed on 1k data instead of 10k for CIFAR-10 and 5k for ImageNet, which reported in the paper.
5. Although the paper proposes an interesting idea, the evaluation is unreliable. Since random factors (Gaussian noise) are introduced in the input layer, gradient-based attacks may fail. The method should be re-evaluated using AutoAttack within the rand version.
6. The paper lacks comparison with more advanced methods, such as diffusion-based AP methods.

**Questions:**

See Weaknesses.

---

> ### Author Response · Authors · 2025-11-21
>
> The results in Table 2 are confusing. All accuracy are shown as xx.x0%. It seems that the evaluation is performed on 1k data instead of 10k for CIFAR-10 and 5k for ImageNet, which reported in the paper.
>
> Please refer to lines 317-321. In fact, "Diffusion models for adversarial purification," ICML 2022 only tests 512 images.
>
> Although the paper proposes an interesting idea, the evaluation is unreliable. Since random factors (Gaussian noise) are introduced in the input layer, gradient-based attacks may fail. The method should be re-evaluated using AutoAttack within the rand version.
>
> The random version of Autoattack only consists of APGD-CE and APGD-DLR with EOT-20. With our random model, we carefully identify that the higher robust accuracy against APGD-$l_2,_\infty$ than PGD through different BPDA options, so we choose PGD. Moreover, we also test EOT-50 instead of only EOT-20. Please refer to lines 366-369.
>
> The paper lacks comparison with more advanced methods, such as diffusion-based AP methods.
>
> Please refer to lines 139-145.
>
> For other comments about the writing, thanks.

---

> > ### Comment · Reviewer_NjGZ · 2025-11-26
> >
> > Thank you for your rebuttal. Some of my concerns have been addressed, so I am raising my rating to 4. However, as the overall quality of the paper remains below average, I still lean toward a borderline rejection.

---

### Official Review · Reviewer_hGLu · 2025-10-28

**Soundness:** 3
**Presentation:** 3
**Contribution:** 3
**Rating:** 6
**Confidence:** 4

**Summary:**

This paper proposes an input obfuscating layer to defend against unseen $l_p$ bounded adversarial attacks. The obfuscating layer is simplified with a sign function during test time. The paper shows the effectiveness with WideResNet and ConvNeXT on CIFAR-10 and ImageNet-1k.

**Strengths:**

- The proposed method is computationally efficient. Its training is almost as same as standard training with clean images.
- The proposed method shows effectiveness against different attacks while considering obfuscated gradients.
- The proposed method also shows effectiveness against unlearnable examples.
- The paper provides the difference between test-time input transformation and training-time one as ablation studies.
- The paper explains the importance of randomness in its method by using a deterministic model.

**Weaknesses:**

- Although the proposed defense is computationally efficient, there is about 5% clean accuracy drop and lower robust accuracy, especially for $l_2$ and $l_\infty$ compared to adversarial training on ImageNet-1k.
- Although BPDA is embedded by default in the evaluation, the reviewer thinks adaptive attacks should be discussed.
- By design, existing adversarial attacks are ineffective with a small noise budget due to extreme squeezing. There are no results with higher adversarial budgets. The reviewer thinks that if using even 8/255 instead of 4/255, the existing attacks can be more successful.

Minor:
- The writing can be improved. In its current form, the paper uses more spoken English. The community may prefer a formal tone in the paper.
- In the abstract, 100/50 may appear to be 100 divided by 50. It is better to say 100 or 50 epochs.
- In Table 3, Clean and BPDA are a bit confusing, although Clean is for the column and BPDA is for the row. It is better to restructure the tables to be clearer.
- In the caption of Table 4, there should be a space between training and (N.T).
- Please take care of all punctuation.

**Questions:**

- In Table 4, are the training transformation and the test transformation in the evaluation the same? N.T trained models for N.T in attacks and T.T for T.T? Have you tried the attacks in N.T and then the resulting adversarial examples in T.T? If so, what is the performance difference?

- Have you tried the proposed defense on bigger models? The reviewer wonders what feature squeezing will do to bigger models. How about vision transformers?

---

> ### Author Response · Authors · 2025-11-21
>
> Thank you very much for your support.
>
> Although the proposed defense is computationally efficient, there is about 5% clean accuracy drop and lower robust accuracy, especially for $l_2$ and $l_\infty$ compared to adversarial training on ImageNet-1k.
>
> When tested against the powerful DDN adversarial attack (which uses the L2 norm), our defense method remains effective and robust. In contrast, the competitors are broken by this same attack.
>
> Please refer to Table 9.
>
>
>
> In Table 4, are the training transformation and the test transformation in the evaluation the same? N.T trained models for N.T in attacks and T.T for T.T? Have you tried the attacks in N.T and then the resulting adversarial examples in T.T? If so, what is the performance difference?
>
> In our approach, the test network remains fixed. It incorporates additional noise and a sign component compared to the standard network, as illustrated in the lower section of Figure 2. This design choice significantly streamlines the evaluation process. We explore two distinct training schemes with different network configurations: Normal Training (N.T.), also referred to as Detour Training (shown in the upper part of Figure 2), and Training within the Test framework (T.T.). While it is feasible to perform transfer attacks by crafting adversarial samples from a training framework, such attempts prove ineffective in our setup, as detailed in lines 428-431.
>
> Regarding the adaptive attack, as the only extra components are noise and a sign component, we have explored all possibilities to verify that ours is a genuine defense, i.e., different BPDA and EOT.
>
>
>
> Have you tried the proposed defense on bigger models? The reviewer wonders what feature squeezing will do to bigger models. How about vision transformers?
>
> Thank you for this excellent suggestion. Unfortunately, due to resource limitations, we have not been able to try this approach.
>
>
> There are no results with higher adversarial budgets. The reviewer thinks that if using even 8/255 instead of 4/255, the existing attacks can be more successful.
>
> Yes, you are right.
>
> | Dataset  | PGD-$l_\infty$ 8/255   | PGD-$l_\infty$ 16/255 |
> | -------- | :-----------------------------------------------------------------------------: | :------------------------------------------------------------------------------: |
> | CIFAR-10 |                                      42.38                                      |                                      11.40                                       |
>
>
> | Dataset  | PGD-$l_\infty$ 4/255   | PGD-$l_\infty$ 8/255 |
> | -------- | :-----------------------------------------------------------------------------: | :------------------------------------------------------------------------------: |
> | ImageNet |                                      24.18                                      |                                       2.82                                       |
>
> ------------------------------------
>
> Thank you for all your writing suggestions.

---

> > ### Comment · Reviewer_hGLu · 2025-11-24
> >
> > Thank you for your effort and answering the questions.The reviewer especially appreciates the effort towards BPDA and EOT.
> >
> > Even though good gradients are estimated for the noise and sign components, the proposed defense is designed to randomly drop changes in the pixel at every inference. So, the reviewer thinks that the adaptive attacker has to think differently from BPDA and EOT.
> >
> > For example, one idea might be to observe whether there are common pixels that do not get dropped for multiple inferences. If such pixels exist, an adaptive attack should modify only those pixels. The changes by the attack have to reflect the input image after the sign function. The reviewer sincerely thinks that a small section of adaptive attacks that discusses new ideas will be interesting.

---

> ### Author Response · Authors · 2025-11-27
>
> Thank you very much for your response.
>
> As requested, we have conducted the following experiments. For CIFAR-10 and ImageNet, we selected 1000 correctly classified test samples and identified the common elements that consistently remained as 1 or 0 after noise and sign processing across 5 runs. We then attempted to perturb these elements using the PGD-$l_\infty$ 8/255 for CIFAR-10 and PGD-$l_\infty$ 4/255 for ImageNet.
>
> In the following table, we show some statistics of the results.
> For example, in the first sample (row) of the Table for CIFAR-10, the number of elements with value 1 is 436, while the number of elements with value 0 is 827. Given that the total input dimension is 32×32×3 = 3072, the total number of elements targeted for flipping is 436 + 827 = 1263. Under the attack, 213 of these elements were successfully flipped, yet the sample remained correctly classified.
>
> Please note that test sample ID 4 was initially misclassified, so it was not subjected to this attack.
>
>
>
> | test ID of CIFAR-10 | Elements 1            | Elements 0            | Successfully Flipped | Classification  |
> |---------|----------------------|----------------------|---------------------|-----------------|
> | 0       | 436/3072 (14.19%)  | 827/3072 (26.92%)  | 213/1263 (16.86%)   | correct         |
> | 1       | 1541/3072 (50.16%) | 608/3072 (19.79%)  | 97/2149 (4.51%)     | correct         |
> | 2       | 1026/3072 (33.40%) | 739/3072 (24.06%)  | 165/1765 (9.35%)    | correct         |
> | 3       | 1551/3072 (50.49%) | 444/3072 (14.45%)  | 186/1995 (9.32%)    | correct         |
> | 5       | 281/3072 (9.15%)   | 1043/3072 (33.95%) | 223/1324 (16.84%)   | correct         |
>
> | test ID of ImageNet | Elements 1               | Elements 0               | Successfully fliped  | Classification  |
> |---------|-------------------------|-------------------------|----------------------|-----------------|
> | 0       | 17762/150528 (11.80%) | 35148/150528 (23.35%) | 25850/52910 (48.86%) | correct         |
> | 1       | 70487/150528 (46.83%) | 11901/150528 (7.91%)  | 26360/82388 (31.99%) | correct         |
> | 3       | 11580/150528 (7.69%)  | 57171/150528 (37.98%) | 34284/68751 (49.87%) | correct         |
> | 5       | 19406/150528 (12.89%) | 19165/150528 (12.73%) | 25017/38571 (64.86%) | correct         |
> | 6       | 21945/150528 (14.58%) | 12906/150528 (8.57%)  | 23335/34851 (66.96%) | correct         |
>
> Evaluation on 1000 initially correctly classified samples reveals a robust accuracy under attack of 97.30% for CIFAR-10 and 98.10% for ImageNet.
>
> The reviewer sincerely thinks that a small section of adaptive attacks that discusses new ideas will be interesting.
>
> Sure. We will do that.

---

> > ### Comment · Reviewer_hGLu · 2025-11-27
> >
> > Thank you for your new experiments and explanation. I increase the score.

---

> > > ### Author Response · Authors · 2025-11-27
> > >
> > > Thank you very much for your support.

---

### Official Review · Reviewer_tA7Z · 2025-10-29

**Soundness:** 3
**Presentation:** 3
**Contribution:** 3
**Rating:** 4
**Confidence:** 2

**Summary:**

The paper proposes a randomized smoothing method which works as an input layer and defends against both unseen and one pixel attacks. The proposed defense does not utilize any prior knowledge from the attacker ends.

**Strengths:**

1. Compared to competitive methods the proposed method does not require any adversarial training which is a general and efficient way of defending against adversarial attacks.

2. This is a lightweight defense with minimum additional parameters and no extra training cost.

3. The proposed method is grounded by proper theoretical analysis and the impact of defending against unseen attack is high.

**Weaknesses:**

Evaluation of Attacks: While the attack claims there  that two prior works can defend against  adversarial training and in the evaluation their performance against the BPDA attack on l_infinity is higher the proposed defense. My concern is that in that case they outperform the proposed method. Even though the authors argue that adversarial training is costly but now it is possible to achieve adversarial training for free (https://proceedings.neurips.cc/paper_files/paper/2019/file/7503cfacd12053d309b6bed5c89de212-Paper.pdf).

Model architecture: Evaluation on conventional model architecture is great, but the paper would benefit from evaluating on recent ViT architecture or even VLM evaluation. Since this should defend against unseen attack why not test on Object detection model  attacks as well.

**Questions:**

Please refer to the weakness section.

---

> ### Author Response · Authors · 2025-11-21
>
> My concern is that in that case, they outperform the proposed method.
>
> The competitors indeed achieve higher robust accuracy under an $l_\infty$ threat model. However, we would like to emphasize that, since our method uses only standard training on clean images, a fair comparison should be restricted to models trained in the same manner. As is well established in the literature, standardly trained models typically exhibit zero robust accuracy against such adversarial attacks.
>
> Even though the authors argue that adversarial training is costly but now it is possible to achieve adversarial training for free.
>
> This method is not applicable in unseen attack scenarios. Furthermore, it was published in 2019, thereby preceding all of our competitors, whose works date from 2021 to 2023. This chronological gap strongly evidences its limited relevance to the current research landscape.
>
> Model architecture: ..
> Thanks.

---

### Official Review · Reviewer_fuWq · 2025-11-01

**Soundness:** 3
**Presentation:** 2
**Contribution:** 3
**Rating:** 6
**Confidence:** 3

**Summary:**

The paper proposes a new defense mechanism, Randomized Feature Squeezing (RFS), that enhances adversarial robustness without using adversarial training. The approach introduces a specially designed input layer that performs randomized feature squeezing through reciprocal and multiplicative transformations combined with Gaussian noise. This layer is trained end-to-end on clean data and removed during inference, leaving a simplified test framework. Experiments on CIFAR-10 and ImageNet show strong robustness against unseen attacks, as well as unlearnable examples (OPS), while maintaining reasonable clean accuracy and lower computational cost compared to adversarial training methods.

**Strengths:**

1. About design. The proposed randomized feature squeezing layer is original and well-motivated, addressing adversarial robustness without adversarial data augmentation.
2. About evaluation. Experiments include both black-box and white-box attacks across multiple norms and datasets, showing consistent and competitive performance.
3. About efficiency. The approach dramatically reduces training cost and time compared to traditional adversarial training, demonstrating practical feasibility.

**Weaknesses:**

1. Potential gradient obfuscation. Despite the authors’ discussion, the reliance on reciprocal and Sigmoid/Sign operations raises concerns about obfuscated gradients, and the robustness might partly stem from non-differentiability rather than genuine defense.
2. Limited theoretical analysis. While empirical performance is strong, there is little formal justification or theoretical insight into why feature squeezing inherently yields robustness across norms.
3. Evaluation scope. Robustness is measured on subsets and with fixed attack budgets, which might limit the generalizability of results. Moreover, it seems that more adversarial attack methods should be evaluated.
4. Ablation insufficiency. Although ablation studies are included, the influence of key hyperparameters (σ, α, kernel size) on both clean and robust accuracy is not deeply explored.
5. Reduced clean accuracy. The clean accuracy, especially on ImageNet, is lower than adversarially trained models, suggesting a trade-off that could be more explicitly quantified.

**Questions:**

1. How do you ensure that the robustness does not primarily result from gradient masking? Could you provide quantitative analysis such as gradient norms or adaptive attack results beyond BPDA and EOT?
2. What is the sensitivity of the model to the noise level σ and sparse loss weight α? Can improper tuning drastically reduce robustness?
3. Would combining RFS with light adversarial or noise-based training further improve clean accuracy without large cost?
4. Can this approach generalize to other architectures (e.g., ViTs) or tasks (e.g., segmentation, detection)?
5. Could you elaborate on how RFS handles real-world attacks where perturbations may not conform strictly to lp norms?

---

> ### Author Response · Authors · 2025-11-21
>
> Thank you very much for your support.
>
> How do you ensure that the robustness does not primarily result from gradient masking?
>
> This question returns to the fundamental problem of why neural networks can generalize well to unseen samples. Existing theories often assume that the probability distribution functions of the test and training sets are identical, i.e., $P_{\text{test}}(x) = P_{\text{train}}(x)$. However, a critical issue arises: given a test image $x$, what is $P_{\text{train}}(x)$?
>
> If $P_{\text{train}}(x)$ is high, we could confidently assume that $x$ will be classified correctly even without performing inference. Unfortunately, $P_{\text{train}}(x)$ is generally inaccessible in practice. As a result, we rely on high test accuracy to build confidence that the network has been well-trained—though this approach lacks scientific rigor.
>
> For instance, if we replace the entire test set with adversarial samples, the test accuracy drops to zero. Surprisingly, these adversarial samples often closely resemble natural images, implying that $P_{\text{train}}(x)$ should intuitively be high—yet in reality, it is not. This discrepancy stems from the vastness of the input space $\mathcal{X}$. More specifically, the region in which the network generalizes well, denoted as $S_g \subseteq \mathcal{X}$, may not align with human perceptual space, denoted as $S_a \subseteq \mathcal{X}$. Both $S_g$ and $S_a$ occupy certain portions of the input space, which is so expansive that $S_g$ and $S_a$ may have little or no overlap, i.e., $S_g \cap S_a = \emptyset$.
>
> We therefore propose that by severely constraining the input space, a well-trained network—which inherently possesses some generalization capacity—is compelled to align its region of good generalization $S_g$ more closely with human perceptual space $S_a$. This forced alignment in a reduced space directly translates to superior adversarial robustness, as conclusively demonstrated by our experiments.
>
> Indeed, under deterministic models, a deep connection exists between feature squeezing and Rademacher complexity for adversarially robust generalization in [1]. Let $x \in X \subseteq \mathbb{R}^d$ and $y \in Y \subseteq \{-1,1\}$ be the feature and label spaces, and the training set $\{(x_1,y_1),(x_2,y_2),\dots,(x_n,y_n)\}$. The experimental Rademacher complexity for the function class $F$ is defined as:
>
> $R_s(F) = \frac{1}{n} \mathbb{E}\_\sigma \left[ \sup_{f \in F} \sum_{i=1}^n \sigma_i f(x_i) \right]$
>
> where $\sigma_1, \sigma_2, \dots, \sigma_n$ are independent random variables uniformly chosen from $\{-1,1\}$, and $\mathbb{E}_\sigma$ denotes the expectation.
>
> The main result is as follows:
>
> ---
>
> **Theorem**
> Let $F := \{ \langle w, x \rangle : \|w\|_p \le W \}$
>
> and
>  $\tilde{F} := \{ \min_{x' \in B_x^\infty(\varepsilon)} y \langle w, x' \rangle : \|w\|_p \le W \}$.
>
>  Suppose that $\frac{1}{p} + \frac{1}{q} = 1$. Then there exists a universal constant $c \in (0,1)$ such that
>
> $
> \frac{c}{2} \left( R_s(F) + \varepsilon W \frac{d^{1/q}}{\sqrt{n}} \right) \le R_s(\tilde{F}) \le R_s(F) + \varepsilon W \frac{d^{1/q}}{\sqrt{n}}.
> $
>
> ---
>
> It indicates that experimental Rademacher complexity $R_s(\tilde{F})$ for adversarial settings depends on the input dimension, and thus affects the generalization ability of the adversarial learning algorithm. We reduce the input dimension by transforming the pixels of the input image to the extreme values of 0 and 1, so the test adversarial accuracy should not be far from the training one. That is verified by our experiments, with a maximum discrepancy of less than 10 in most cases, as shown in Table 5.
>
> This theoretical justification is not rigorous; however, it certainly helps to prove that our approach is a genuine defense.
>
> [1] Dong Yin, Kannan Ramchandran, and Peter L. Bartlett. Rademacher complexity for adversarially robust generalization. ICML 2019
>
>
> Regarding the adaptive attacks, we have exhausted all existing ones. Based on the streamlined nature of our testing framework and the above arguments, we are highly confident in the robustness and effectiveness of our proposed method.
>
> Evaluation scope. Robustness is measured on subsets and with fixed attack budgets, which might limit the generalizability of results. Moreover, it seems that more adversarial attack methods should be evaluated.
>
> We have also done experiments on DDN, C&W, and Spatial Transform attacks, which are optimization-based approaches without fixed attack budgets, and ours is the only one that can defend against DDN. Please refer to Table 9.

---

> ### Author Response · Authors · 2025-11-21
>
> Additional ablation studies are presented in the following tables, with the bolded ones corresponding to the models used in our paper. There is no significant difference between the different choices.
>
> For ImageNet:
>
> | kernel_size | $\alpha$        |  $\sigma$       | Clean  | PGD-$l_\infty$  |
> | ----------- | ----------- | ----------- | :-----------------------: | :----------------------: |
> | **7x7** | **0.5** | **1.4** |       **67.80** |      **23.74** |
> | 5x5         | 0.5         | 1.4         |           67.30           |          25.20           |
> | 9x9         | 0.5         | 1.4         |           66.94           |          24.06           |
> | 3x3         | 0.5         | 1.2         |           68.04           |          22.24           |
> | 5x5         | 0.5         | 1.2         |           68.82           |          22.84           |
>
> For CIFAR-10:
> | kernel_size | $\alpha$      | $\sigma$      | Clean   | PGD-$l_\infty$  | Nattack Determinstic N = 5/20/30 |
> | ----------- | ---------- | ----------- | :------------------------------------------: | :----------------------------------------: | :---------------------------------------------------------: |
> | **5x5** | **0.1** | **0.65** | **80.14** | **43.30** | **19.00/28.00/28.00** |
> | 5x5         | 0.1        | 0.6         | 80.92                                      | 40.40                                    |                                                           |
> | 5x5         | 0.1        | 0.7         | 79.50                                      | 42.20                                    | 20.00/25.00/26.00                                         |
> | 5x5         | 0.2        | 0.7         | 79.69                                      | 46.80                                    |                                                           |
> | 5x5         | 0.3        | 0.7         | 78.99                                      | 45.20                                    |                                                           |
> | 5x5         | 0.4        | 0.7         | 68.97                                      | 42.00                                    |                                                           |
> | 5x5         | 0.5        | 0.7         | 60.34                                      | 47.20                                    |                                                           |
> | 7x7         | 0.2        | 0.7         | 79.71                                      | 51.90                                    |                                                           |
> | 9x9         | 0.2        | 0.7         | 79.94                                      | 55.80                                    | 19.00/24.00/25.00                                         |
> | 11x11       | 0.2        | 0.7         | 79.83                                      | 53.30                                    |                                                           |
>
>
>
> Our initial experiments indicated that a 9x9 kernel yielded the highest robust accuracy for CIFAR-10 under a PGD-$l_\infty$ white-box attack. However, a more comprehensive evaluation revealed that these models did not maintain their superiority under the gradient-free NAttack for the deterministic models. Consequently, we adopt a rigorously conservative evaluation framework to ensure the sustainability of our results.
>
> Would combining RFS with light adversarial or noise-based training further improve clean accuracy without a large cost?
>
> No. The strong feature compression inherent in our method leads to an inevitable decrease in clean accuracy, reflecting the well-known trade-off between robustness and standard performance.
>
> Can this approach generalize to other architectures (e.g., ViTs) or tasks (e.g., segmentation, detection)?
> Could you elaborate on how RFS handles real-world attacks where perturbations may not conform strictly to lp norms?
>
> Thank you for your suggestion. We will explore these in future work.

---

### Meta-Review · Area_Chair_hci3 · 2026-01-03

**Summary:**

The four reviewers’ comments are mixed but converge on several key concerns, including:
(1) the lack of a formal justification for why the proposed Randomized Feature Squeezing method improves robustness;
(2) the limited scope of experimental evaluation, particularly with respect to the datasets, attack budgets, attack types, and model architectures considered; and
(3) the observed drop in clean accuracy and inferior robust accuracy compared with adversarial training on ImageNet-1k.

In addition, the reviewers raise several questions regarding technical details and presentation clarity.

AC comments. The AC also examined the paper at certain length and found that several claims made by the authors are either insufficiently supported or factually incorrect. For instance, the claimed second contribution—“our work is the only one that does not require any prior knowledge about the attacks, using standard training with clean images while achieving strong robustness”—is not accurate.

First, the claim is both overly broad and vague. Second, prior work exists that improves both clean accuracy and robustness using standard training on clean data, without assuming any prior knowledge of adversarial attacks. While these works may not explicitly frame their contributions from the perspective of adversarial robustness, they nevertheless contradict the authors’ exclusivity claim. Representative examples include:

[1] E.-H. Yang et al., “Conditional Mutual Information Constrained Deep Learning for Classification,” IEEE Transactions on Neural Networks and Learning Systems, vol. 36, no. 8, Aug. 2025.
[2] A. H. Salamah et al., “JPEG-Inspired Deep Learning,” ICLR 2025.

**Reviewer Concerns:**

Some of the concerns raised by Reviewers hGLu and NjGZ were adequately addressed in the rebuttal. Major concerns are still outstanding. In general, the rebuttal does not provide full responses to questions and concerns raised by reviewers.

**Reviewer Scores:**

Reviewers hGLu and NjGZ indicated to increase their respective scores slightly, but Reviewer NjGZ remains negative. The work is not mature enough. The authors are encouraged to continue to work on it.

---

### Decision · Program_Chairs · 2026-01-26

Reject